# Serpentovirus (Nidovirus) and Orthoreovirus Coinfection in Captive Veiled Chameleons (*Chamaeleo calyptratus*) with Respiratory Disease

**DOI:** 10.3390/v12111329

**Published:** 2020-11-19

**Authors:** Laura L. Hoon-Hanks, Anke C. Stöhr, Amanda J. Anderson, Dawn E. Evans, Javier G. Nevarez, Raúl E. Díaz, Case P. Rodgers, Shaun T. Cross, Halley R. Steiner, Roy R. Parker, Mark D. Stenglein

**Affiliations:** 1Department of Microbiology, Immunology, and Pathology, College of Veterinary Medicine and Biomedical Sciences, Colorado State University, Fort Collins, CO 80523, USA; laura.hoon-hanks@colostate.edu (L.L.H.-H.); cprodger@rams.colostate.edu (C.P.R.); shaun.cross@colostate.edu (S.T.C.); 2Department of Veterinary Clinical Sciences, School of Veterinary Medicine, Louisiana State University, Baton Rouge, LA 70803, USA; anke.stoehr@freenet.de (A.C.S.); jnevare@lsu.edu (J.G.N.); 3Louisiana Animal Disease Diagnostic Laboratory, Baton Rouge, LA 70803, USA; aanderson@cvm.tamu.edu (A.J.A.); devans1@lsu.edu (D.E.E.); 4Department of Biological Sciences, California State University, Los Angeles, CA 90032, USA; Lissamphibia@gmail.com; 5Department of Biochemistry, University of Colorado, Boulder, CO 80303, USA; halley.r.steiner@colorado.edu (H.R.S.); roy.parker@colorado.edu (R.R.P.)

**Keywords:** nidovirus, serpentovirus, respiratory disease, chameleon, reptile, reovirus

## Abstract

Serpentoviruses are an emerging group of nidoviruses known to cause respiratory disease in snakes and have been associated with disease in other non-avian reptile species (lizards and turtles). This study describes multiple episodes of respiratory disease-associated mortalities in a collection of juvenile veiled chameleons (*Chamaeleo calyptratus*). Histopathologic lesions included rhinitis and interstitial pneumonia with epithelial proliferation and abundant mucus. Metagenomic sequencing detected coinfection with two novel serpentoviruses and a novel orthoreovirus. Veiled chameleon serpentoviruses are most closely related to serpentoviruses identified in snakes, lizards, and turtles (approximately 40–50% nucleotide and amino acid identity of ORF1b). Veiled chameleon orthoreovirus is most closely related to reptilian orthoreoviruses identified in snakes (approximately 80–90% nucleotide and amino acid identity of the RNA-dependent RNA polymerase). A high prevalence of serpentovirus infection (>80%) was found in clinically healthy subadult and adult veiled chameleons, suggesting the potential for chronic subclinical carriers. Juvenile veiled chameleons typically exhibited a more rapid progression compared to subadults and adults, indicating a possible age association with morbidity and mortality. This is the first description of a serpentovirus infection in any chameleon species. A causal relationship between serpentovirus infection and respiratory disease in chameleons is suspected. The significance of orthoreovirus coinfection remains unknown.

## 1. Introduction

A number of viruses cause disease in captive and wild reptiles, and new viral pathogens are continuously being discovered throughout the world [1,2,3]. In 2014, a novel group of nidoviruses (order *Nidovirales*), currently classified as serpentoviruses, were identified as potential reptile pathogens [4,5,6,7,8]. The first descriptions of serpentovirus-associated disease were in ball pythons (*Python regius*) and Indian pythons (*Python molurus*), shortly followed by green tree pythons (*Morelia viridis*) [4,5,6,7]. In these cases, snakes were dying of interstitial proliferative and mucinous pneumonia, tracheitis, and esophagitis. In 2016, a closely related serpentovirus was also discovered in wild shingleback lizards in Australia [9], marking the first description of a serpentovirus in a reptile outside the Ophidia clade. Some infected shinglebacks showed clinical evidence of respiratory disease similar to that described in snakes. However, further evidence of disease causation was not explored. Subsequently, related serpentoviruses have been discovered in additional snake species, turtles, with or without disease association [10,11,12,13,14,15,16,17]. Serpentoviruses are now regarded as significant respiratory pathogens of snakes, but their association with disease in other reptile species is less well established.

The following report describes an outbreak of respiratory disease in veiled chameleons (*Chamaeleo calyptratus*). Aspects of the case history were suggestive of a possible infectious etiology, but traditional diagnostic methods did not reveal a potential bacterial, fungal, or viral agent. Using metagenomic sequencing, two novel serpentoviruses (nidovirus) and a reptilian orthoreovirus coinfection were identified in association with respiratory disease in the chameleons. Furthermore, the histologic, genomic, and phylogenetic characterization of these novel viruses was performed. The findings in this study highlight the significance of emerging viruses in the health of captive reptile populations and provide additional support for the hypothesis that serpentoviruses are significant respiratory pathogens of a variety of non-avian reptiles.

## 2. Materials and Methods 

### 2.1. Ethics Statement

The collection of lizards described below was enrolled in a separate biological research project for which IACUC review was independently completed (Southeastern Louisiana University IACUC Protocol #0046; approval 26-Oct-2017). For the study described in this report, all antemortem samples were considered non-invasive clinical samples collected by veterinarians for diagnostic purposes. Postmortem samples were also collected for diagnostic purposes. Euthanasia of animals was performed by a veterinarian for the purposes of herd health; use of diagnostic samples for this study was considered secondary and did not influence clinical decisions. For these reasons an ethical review for this study was not performed.

### 2.2. Case History

Captive-bred veiled chameleons (VC) were purchased from different commercial breeding facilities in the United States over the course of 13 months as part of a biological research project (Figure 1). Husbandry practices at the research facility were as follows: chameleons were housed in a room with 30–50% humidity. The enclosures were 18″ × 18″ × 36″ screen mesh enclosure with no substrate on the bottom. Juveniles (1–5 months old) and small subadults (5–12 months; less than 75 g) were housed 2–3 per enclosure while large subadults (greater than 75 g) and adults (greater than 12 months) were housed singly. The enclosures had a UVB bulb (Repti-Glo 5.0 Fluorescent Lamp, 40 W, 48 inches, Exo Terra©, Mansfield, MA, USA replaced regularly per manufacturer’s recommendation), and a heat lamp. The temperature in the enclosure was a minimum of 75 °F with a maximum basking area of 90–100 °F. The animals were sprayed with reverse osmosis-filtered water 2–4 times per day. The diet consisted of crickets dusted with calcium (Calcium powder, Rep-Cal©, Los Gatos, CA, USA) 3–4 times per week, calcium with vitamin D3 (Calcium with Vitamin D3 powder, Rep-Cal©) twice a month, as well as dusted with vitamin and mineral powder (Herptivite Multivitamin, Rep-Cal©) two times a month. Occasionally, mealworms were offered [18].

In September 2017, eight captive-bred juvenile veiled chameleons were purchased from a commercial breeding facility. Within 3–4 weeks of arriving in the research facility, all chameleons began to exhibit respiratory signs: wheezing and vertical head tilting with gasping, increased mucus in the oral cavity, anorexia, and reduced water intake. All juvenile chameleons died within 0.5–1.5 months of arrival (Figure 1). The enclosures and items in the enclosures were washed with soap and water prior to the introduction of new chameleons, but other disinfectants were not used. 

Between October 2017 and January 2018, seventeen additional captive-bred veiled chameleons (4 subadults and 13 juveniles) were obtained from three different U.S. commercial breeders. Shortly after their introduction into the same enclosures as above, juvenile animals exhibited similar signs of respiratory disease and nine animals died; the subadults did not show clinical signs during this time. 

In February 2018, two of the four remaining juveniles from this group, one male with clinical signs of respiratory disease (VC1; 2 months old) and one clinically healthy female (VC2; 2.5 months old), were evaluated by the Louisiana State University Veterinary Teaching Hospital Zoological Medicine Service. On presentation, VC1 weighed 6.3 g, was quiet, alert and responsive, with an appropriate body condition score (BCS 4/9). Increased respiratory effort (respiratory rate 3/min), wheezing and gasping for air were observed. The rest of the physical exam was unremarkable. VC2 weighed 4.6 g, was bright, alert and responsive, in appropriate body condition (BCS 4/9), and had no abnormalities on physical examination. Both animals were euthanized with 0.2 mg pentobarbital sodium (FATAL-PLUS solution, Vortech Pharmaceuticals©, Dearborn, MI, USA) IV in the ventral coccygeal vein and submitted for necropsy to the Louisiana Animal Disease Diagnostic Laboratory. No diagnostics or treatments had been performed prior to this. The case history, clinical findings, and postmortem findings (see below) suggested an infectious agent could be playing a role. Spleen, pancreas, liver, lung, stomach, small and large intestine, heart, ovary/testis, kidney, and head (nasal and oral cavities, brain, and eyes) were placed in 10% neutral buffered formalin for histopathology. Fresh VC1 and VC2 samples were submitted separately for bacterial and parasitic evaluation: liver tissue, lung swabs, and colon contents were submitted for aerobic culture, liver and lung samples were pooled for enriched culture of *Salmonella* species, and feces were submitted for fecal flotation. Fresh-frozen lung from VC1 was submitted for reptile paramyxovirus PCR to the University of Florida ZooMed Diagnostic Laboratory. Fresh-frozen lung, liver, and kidney were collected and stored at −80 °C, and subsequently shipped overnight on dry ice and ice packs to Colorado State University (CSU) for metagenomic pathogen detection.

In the next three weeks (end of February 2018), the remaining two juvenile chameleons also developed clinical signs and died. The whole carcass of one juvenile chameleon (identified as VC3) that died during this time was frozen at −80 °C and also shipped to CSU for metagenomic pathogen detection.

An environmental problem was considered a possible cause or contributor to the respiratory disease. Too high humidity was suspected, resulting in the addition of a dehumidifier in the housing area. Despite this modification mortalities continued (3 subadults). Subsequently, the ceiling vents were completely closed to prevent air exchange and the room temperature was increased to 80 °F, which resulted in no further mortalities; 1 subadult chameleon remained alive at this time (VC4). 

In October 2018, 5 additional veiled chameleons were obtained (1 adult [VC5], 1 subadult [VC6], and 3 juveniles [VC7–9]). These chameleons were housed in the same area as the one remaining chameleon from the previous group (VC4, now an adult), resulting in a total of 6 chameleons (VC4–9). In December 2018, oral/choanal swabs were collected using sterile cotton-tipped swabs and placed in viral transport medium (modified phosphate-buffered sucrose with aminoglycosides) [19]; the samples were chilled directly on ice and then frozen at −80 °C within 4 h of collection. Swabs were shipped to CSU for virological testing.

In February 2019, due to concern over the health of the collection, all remaining chameleons were euthanized (VC4 and 6-9) despite showing no evidence of respiratory disease; VC5 died a few weeks prior after development of a nasal abscess (no necropsy performed). Carcasses were submitted for necropsy to the Louisiana Animal Disease Diagnostic Laboratory. Trachea, lung, kidney, heart, liver, spleen, pancreas, stomach, small and large intestine, gonads, and adrenal glands were placed in 10% neutral buffered formalin for histopathology. Fresh-frozen lung and trachea were pooled together for each animal and shipped to CSU for virological testing.

Additional lizard species in the collection that were temporarily housed in the same room as chameleons (bearded dragons [*Pogono vitticeps*], *n* = 6) or housed in the same facility in a separate room (common leopard geckos [*Eublepharis macularius*], *n* = 3; ocelot geckos [*Paroedura pictus*] n = 3) were clinically healthy throughout this period. Choanal swabs from these animals were also collected in December 2018 for testing at CSU.

### 2.3. Metagenomic Sequencing and Data Analysis

Total RNA was extracted from fresh-frozen lung, liver, and kidney pools (VC1 and VC2) and oral mucosa, trachea, and lung pool (VC3) using a combination of TRIzol (Ambion Life Technologies, Carlsbad, CA, USA) with RNA clean and concentrator columns (CC-5; Zymo Research, Irvine, CA, USA) as previously described [15]. RNA libraries were generated using the Kapa RNA HyperPrep Kit (Kapa Biosystems, Wilmington, MA, USA) according to the manufacturer’s instructions with an input concentration of approximately 50–100 ng of RNA and 6–10 rounds of amplification. Kapa Dual-Indexed Adapter Kit Illumina Platform (Kapa Biosystems) was used for adapter ligation and barcoding. Equivalent masses of DNA from each sample were pooled and a subset of libraries were size selected (250–500 basepairs [bp], including Illumina adapters) by gel electrophoresis on a 2% agarose gel (Sage Science Pippin instrument according to the manufacturer’s instructions). Library quantification was performed with the Kapa Biosystems Illumina library quantification kit according to the manufacturer’s instructions. The RNA-seq libraries were sequenced using a dual-indexed, single-end, 1 × 150 method on an Illumina NextSeq 500 instrument with either a NextSeq 500/550 Mid or High Output Kit v2 (150 cycles). An independent metagenomic sequencing run was also performed on the same RNA samples using a library preparation method with enrichment of double-stranded RNA [20]. This was run on the same platform using a dual-indexed, paired-end, 2 × 75 sequencing method. 

Data analysis was performed as previously described [15]. Briefly, adaptor sequences, low-quality bases, and short sequences (<80 bp) were removed (Cutadapt version 1.18). Reads with 99% global pairwise identity were collapsed, leaving only unique reads (CD-HIT-DUP) [21]. Chameleon-derived sequences were removed from the data set by mapping reads with bowtie2 (v2.3.5.1) to the green anole genome (*Anolis carolinensis* AnoCar2.0), leaving non-host sequences for downstream analysis (the green anole genome was the most closely related available reference genome) [22]. Contiguous sequences (contigs) were assembled (SPAdes genome assembler) and both contigs and non-assembled reads were taxonomically categorized as nucleotide and translated sequences using the NCBI nt and nr databases; BLASTn (version 2.9.0+), Diamond (version 0.9.24), GSNAP (version 2018-07-04) alignment tools were utilized [23,24,25,26].

Sequences of interest detected by metagenomic sequencing were utilized for phylogenetic analysis. Nucleotide and protein sequences were aligned using MAFFT software with default parameters (E-INS-i algorithm, 200PAM/k = 2 scoring matrix, gap open penalty of 3, and offset value of 0) in Geneious 11.0.4 [27,28]. Phylogenetic trees were generated using the protein alignments with PhyML (3.3.20180621) in Geneious: Le Gascuel (LG) substitution model, with 1000 bootstrap replicates, an estimated transition/transversion rate, a fixed proportion of invariable sites (0), 4 substitution rate categories, and a fixed gamma distribution parameter (1) [29].

### 2.4. PCR Analysis for Viral RNA

RNA was extracted from oral swabs using a Zymo Research viral RNA kit. Approximately 200 µL was processed according to the manufacturer’s instructions. RNA was eluted in 30 µL of RNase/DNase-free water. RNA from oral swabs and tissue pools were reverse transcribed into complementary DNA (cDNA) for PCR as follows. Five microliters of RNA was added to 200 pmol of a random pentadecamer oligonucleotide (5′ NNNNNNNNNNNNNNN 3′) and incubated for 5 min at 65 °C. A reverse transcription reaction mixture containing 1 × SuperScript III FS reaction buffer (Invitrogen, Carlsbad, CA, USA), 5 mM dithiothreitol (Invitrogen), 1 mM each deoxynucleoside triphosphates (dNTPs), and 100 U SuperScript III reverse transcriptase enzyme (Invitrogen) was added to the RNA-oligomer mix (12 μL total reaction volume) and incubated for 30 min at 42 °C, then 30 min at 50 °C, then 15 min at 70 °C. cDNA was stored at −20 °C.

Primers were designed to amplify a conserved portion of the RNA-dependent RNA polymerase (RdRp) region of ORF1b of serpentovirus sequences: MDS-1518F 5′ TACACCTACTTTCAAGGMGA 3′ and MDS-1519R 5′ GTTGTWGCATCACCASWGGA 3′. PCR was performed using Luna Universal qPCR Master Mix. Twelve microliter reactions included a final concentration of 1× Luna Universal Master Mix and 0.4 µM of each primer mixed with 5 µL of cDNA diluted 1:10 in water. Reaction mixtures were run with the following cycle parameters: 95 °C for 1 min; 95 °C for 15 s and 60 °C for 60 s with 45 cycles; and a melting curve. PCR products were run on a 1.5% agarose gel with ethidium bromide for confirmation of amplification and assessment of amplicon size (expected 752 bp). DNA bands were gel extracted (Zymo Gel DNA Recovery Kit, Irvine, CA, USA) according to the manufacturer’s instructions. Bands were Sanger sequenced by GENEWIZ (San Diego, CA, USA) using the forward primer. Sanger sequencing of PCR amplicons, yielded 708 bp sequences following removal of poor-quality base calls and primer sequences. These were aligned in Geneious 11.0.4 for assessment of percentage nucleotide identity. 

RNA from tissues and oral swabs were also reverse transcribed by a second method to target dsRNA. The method of reverse transcription was as previously described (see first paragraph of this section) with replacement of the initial 65 °C incubation (RNA plus random pentadecamer oligonucleotide) with incubation at 95 °C for 5 min to allow for denaturation of double stranded RNA viruses and better yield of reovirus cDNA. PCR was performed as described above with broad orthoreovirus primers [30] as well as primers designed specifically to the polymerase gene of the orthoreovirus detected in this study: MDS-1579F 5′ CGTCGGGTAGTGCTGTGATT 3′ and MDS-1580R 5′ TAGGGTGCCTGCTCACATTG 3′ as forward and reverse primers, respectively. Thermocycler parameters were as follows: 95 °C for 1 min; 95 °C for 1 min, 47 °C for 1 min, and 72 °C for 1 min with 45 cycles; and 72 °C for 5 min.

### 2.5. Virus Isolation 

Four cell lines were used for virus isolation attempts with fresh-frozen tissue homogenates from VC3 (no tissue remained from VC1–2 for virus isolation attempts): JK cells (boa constrictor kidney) [31], DPHt cells (diamond python heart) [15], IgH2 cells (iguana heart; ATCC, CCCL-108), and VH2 (viper heart; ATCC, CCL-140). Pooled tissues (oral mucosa, lung, trachea) from VC3 were homogenized in brain heart infusion (Becton Dickinson) by manual disruption with a plastic sterile pestle in a 1.5 mL Eppendorf tube. The homogenate was filtered (Merck Millipore UltraFree-MC 0.22 μm centrifugal filter) and 40 μL was inoculated onto JK, DPHt, IgH2, and VH2 cells at 80% cell confluence in a 6-well tissue-culture plate. Each cell type had 2 devoted wells in the plate: one inoculated with VC3 tissue homogenate and 1 sham inoculated with BHI. Cells were maintained in 2 mL of complete cell medium (MEM/EBSS [HyClone], 10% irradiated FBS [HyClone], 10% Nu-Serum1 [Corning], and 2× penicillin-streptomycin solution [HyClone]) and incubated at 30 °C with 5% CO_2_. Approximately half the volume of supernatant was collected at 24 h post-inoculation and stored at −80 °C; this volume was replaced with fresh medium. Supernatant was similarly collected every 48 h for 13 days. On the final collection, cells were trypsinized (200 μL of 0.25% trypsin applied directly to rinsed cells and incubated for 2 min at 37 °C) and both supernatant and cells were stored at −80 °C. RNA was extracted and PCR was performed as previously described for all cell inoculation samples at all time points.

## 3. Results

### 3.1. Postmortem Findings

Postmortem evaluations of VC1 (juvenile chameleon with clinical evidence of respiratory disease) and VC2 (clinically healthy juvenile chameleon) were performed. Gross examination of VC1 revealed a scant amount of stomach contents and empty small intestines and colon, consistent with anorexia. No other gross lesions were observed. Histopathology of VC1 (Figure 2) revealed severe chronic-active bronchointerstitial pneumonia and tracheitis with proliferative and catarrhal changes, moderate chronic lymphocytic and catarrhal rhinitis, and mild histiocytic and heterophilic colitis. Gross examination of VC2 was unremarkable; histopathologic lesions included mild multifocal heterophilic enterocolitis with focal erosion. 

Traditional diagnostic assays did not yield evidence of possible contributing infectious causes. Paramyxovirus PCR of lung tissue from VC1 was negative. Aerobic culture of liver, lung, and colon and fecal flotation were negative for bacteria and parasites in both VC1 and VC2. *Salmonella* was detected in the liver and lung pool by enrichment culture in VC1 and was found to have broad antibiotic sensitivity; *Salmonella* culture was negative in VC2.

The remaining chameleons within the collection (VC4 and VC6–9) that were euthanized due to concerns of viral spread did not develop respiratory signs although they had tested positive by qRT-PCR for serpentovirus infection prior to culling. Gross examination revealed one chameleon (VC9, 59.4 g juvenile) with mild crusting of the eyes bilaterally and a mild amount of mucus within the oral cavity; one chameleon (VC4, 115 g adult) with a poor body condition score (BCS 2/9), sunken eyes, muscular atrophy, and pale mucous membranes; and one chameleon with rounded liver margins (VC6, 74.8 g subadult); no other gross findings were observed. Histologic findings in the lung were limited to small foci of pneumocyte hypertrophy in faveoli (VC6 and VC8), which was associated with fibrosis and mineralization in one case (VC8, 55.6 g juvenile); no changes were noted in the trachea. The nasal and oral cavities were not assessed histologically. A range of other lesions were identified, including focal xanthomatous mural enteritis with coelomic foreign body (VC4), severe heterophilic enteritis with mural granulomas (VC6), splenic lymphoid hyperplasia (VC6), mild lymphocytic portal hepatitis (VC6), rare mineralization of the tunica intima of large cardiac vessels (VC7; 28.1 g juvenile), and hepatocellular vacuolization (VC4 and VC6–9).

### 3.2. Metagenomic Sequencing Findings

Following a lack of causative agents identified by postmortem diagnostic techniques, metagenomic sequencing was performed as a non-biased approach to infectious disease testing. In the first sequencing run performed, the average number of individual reads per sample was 3 × 10^6^. On average, 91%, 18%, and 9% of sequences remained following adaptor and quality filtering, collapsing to unique reads, and filtration of host-derived sequences, respectively. Sequences of two genotypically distinct veiled chameleon serpentoviruses (VCSV) were detected in VC1 (VCSV-B) and VC3 (coinfection with VCSV-A and -B); no sequences were detected in clinically normal VC2. In the second sequencing run performed (enriching for dsRNA), the average number of paired reads per sample was 24.2 × 10^6^, with an average of 90%, 19%, and 17% sequences remaining after filtration. The same serpentoviruses were detected in VC1 (VCSV-B) and VC3 (VCSV-A and -B); 11 read pairs aligned to VCSV-B in VC2. Additionally, sequences aligning to 10 segments of a novel veiled chameleon orthoreovirus (VCOrV) were detected in all three chameleon samples, with the highest number of reads detected in VC2. Sequencing results for each sample are summarized in Table 1. Assembled genomes are available in GenBank for VCSV-A (MT997160), VCSV-B (MT997159) and VCOrV (MT997161–MT997170, segments L1–3, M1–3, and S1–4, respectively). Raw read files for each chameleon are also available in the short read archive (SRA) for the BioProject PRJNA663098: VC1 (SRR12676605, SAMN16119956), VC2 (SRR12676604, SAMN16119958), and VC3 (SRR12676603, SAMN16119957).

### 3.3. Targeted Viral RNA Detection

Following the discovery of two novel serpentovirus sequences in veiled chameleons, degenerate primers were designed for PCR detection of both virus genotypes. Initial metagenomic sequencing detected serpentovirus sequence in VC1 and VC3, but not in clinically healthy VC2. In contrast, PCR detected serpentovirus nucleic acid in all three chameleons (Figure 3). Sanger sequencing of VC1 and VC2 PCR amplicons targeting the RdRp matched VCSV-B found by metagenomics (>99% identity; Figure 4. The sanger sequencing of VC3 yielded poor-quality sequence due to coinfection, as observed by metagenomics, but the sequence chromatograms revealed double peaks consistent with coinfection by VCSV-A and -B. 

Subsequently, the oral swabs from six additional chameleons (VC4–9), six bearded dragons (BD1–6), three leopard geckos (LG1–3) and three ocelot geckos (OG1–3) were analyzed by PCR for the presence of serpentovirus. Five out of 6 chameleons were positive (VC4, 6–9), whereas all the bearded dragons and geckos were negative (Figure 3). The lowest Cts (highest viral RNA levels) were detected in juvenile animals (VC1–3 and VC7–9) compared to the subadult (VC6) and adult (VC4) (Figure 3); the negative chameleon (VC5) was also an adult. Sanger sequences from VC7–9 aligned to VCSV-A with 100% nucleotide identity (Figure 4). VC4 and VC6 had poorer-quality sequence. VC4 more closely aligned to VCSV-A and VC6 to VCSV-B, but chromatograms suggested possible coinfection. Following euthanasia of serpentovirus-infected veiled chameleons (VC4, 6–9), lung and trachea pools were also tested by PCR, yielding identical results.

Attempts at detecting orthoreovirus nucleic acid by qRT-PCR within chameleon tissues and swabs, even those positive by metagenomic sequencing (VC1–3), were unsuccessful, an occurrence that has been previously reported [32]. Results of VCSV and VCOrV identification by metagenomic sequencing and PCR are summarized in Table 2.

### 3.4. Chameleon Serpentoviruses: Genomic and Phylogenetic Analysis

Coding complete genome sequences were generated for of VCSV-A (31,537 bp) and VCSV-B (36,144 bp). VCSV-B represents the second longest RNA virus discovered to date, behind a 41.1 kb planarian nidovirus [33]. Both VSCV genomes had an overlapping ORF1ab gene with a predicted ribosomal frameshift signal (−1;AAAAAC) followed by a spike protein gene. The 3′ end of VCSV-A contained six open reading frames (ORFs) with an identifiable transmembrane protein, membrane protein, and possible nucleocapsid protein. The 3′ end of VCSV-B contained 4 open reading frames (ORFs) with identifiable transmembrane proteins, membrane protein, and possible nucleocapsid protein. Pairwise alignment of ORF1b revealed 53% nucleotide and 50% amino acid identity between VCSV-A and -B. Nucleotide and protein alignments of ORF1b from VCSV-A to other serpentoviruses resulted in 46–53% nucleotide and 40–52% amino acid identity; ORF1b from VCSV-B exhibited 47–50% nucleotide and 40–48% amino acid identity. A phylogenetic tree of the ORF1b protein alignment (Figure 5) revealed VCSVs closely related to serpentoviruses from snakes, lizards, and turtles. 

### 3.5. Chameleon Orthoreoviruses: Genomic and Phylogenetic Analysis

A complete coding genome of an orthoreovirus was obtained from VC2. The orthoreovirus consisted of 10 segments: 3 long (L) segments, 3 medium (M) segments, and 4 small (S) segments. Nucleotide and protein sequence alignments of each segment using BLASTn and BLASTx revealed highest percent identity to reptilian orthoreovirus sequences isolated from a green bush viper (*Atheris squamigera*) and a ball python (GenBank sequences: GCA_000919495.1 and AY238886 to AY238887, respectively). All segments aligned with 71–87% nucleotide and 81–94% amino acid identity to reptilian orthoreovirus (with greater than 95% query coverage) except M1. Segment M1 was less closely related to reptilian orthoreovirus, with which it only shared 71% nucleotide identity (23% query coverage) and 53% amino acid identity, consistent with it representing a reassortant genotype (96% query coverage; Table 3). A phylogenetic analysis of protein alignments for L3 (RNA-dependent RNA polymerase; RdRp) revealed reptilian orthoreoviruses (bush viper reovirus and reptilian orthoreovirus; isolated from green bush vipers) as the closest evolutionary relation (Figure 6). 

### 3.6. Virus Isolation

Reptile cell lines inoculated with fresh-frozen tissue homogenate did not exhibit cytopathic effects (cell death or syncytial cell formation). Neither serpentovirus nor reovirus RNA was detected by PCR at any time points post-inoculation. However, only VC1, 2, and 3 were confirmed to be infected with orthoreovirus and tissue samples were only available from VC3, which had fewer reovirus-mapping reads than did VC2, suggestive of a reduced viral load.

## 4. Discussion

Novel serpentoviruses have recently been found in snake, lizard, and turtle species associated with respiratory or systemic disease, and a serpentovirus has been confirmed as a cause of respiratory disease in pythons [4,5,6,7,9,11,13,14,15,16]. This study is the first to describe novel serpentoviruses in veiled chameleons with respiratory disease, and only the second to describe serpentoviruses in lizards globally. 

In 2016, a respiratory disease in wild shingleback lizards in Australia was found in association with serpentovirus infection [9]. Clinical signs included excessive oral mucus secretions, serous to mucopurulent discharge from the nares and eyes, pale mucous membranes, sneezing, lethargy, anorexia, depression, and poor body condition. Lizards were tested by antemortem swabbing of the oral cavity; histologic lesions associated with viral infection were not evaluated. In this study of chameleons, clinical signs of respiratory disease were similar to those found in shingleback lizards as well as those described in serpentovirus-infected snakes [4,5,6,15]. Furthermore, histologic lesions in at least one chameleon were like those found in snakes: interstitial proliferative and catarrhal pneumonia, rhinitis, and tracheitis, suggesting a similar pathogenesis [4,5,6,7,15]. 

The clinical history in the chameleon collection indicated a possible age-associated predisposition to morbidity and mortality. Juvenile veiled chameleons exhibited a more rapid onset of clinical signs and death compared to subadults. Furthermore, serpentovirus nucleic acid was most abundant in juveniles (VC1–3 and VC7–9) compared to subadults (VC6) and adults (VC4), and the only negative chameleon (VC5) was an adult. In snakes, a slight positive correlation between older age and likelihood of infection has been identified, but is believed to be due to increased time for potential exposure rather than physiologic changes [16]. In this study, the opposite trend was observed, suggesting that physiologic differences, such as incomplete immune development, could contribute to decreasing susceptibility with increasing age, although this hypothesis will require additional investigation. Other possible factors that could contribute to variation in susceptibility include infectious dose, mode of transmission, concurrent infection or comorbidities, nutritional status, stress response, or other immunologic factors.

Serpentoviruses were detected in chameleons exhibiting clinical signs of respiratory disease with or without histologic correlates, as well as clinically healthy animals lacking microscopic disease within the lower respiratory tract. Serpentovirus infections in shingleback lizards [9] and snakes [16] have also been described in clinically healthy animals. This provides further evidence that disease manifestation following serpentovirus infection is variable. Infection in “healthy” animals could represent virus detection during the incubation period, detection in a non-clinical carrier, or detection in an animal with non-respiratory/systemic infection. In a previous study, some pythons that were consecutively tested for over two years were continuously positive for serpentovirus but never exhibited clinical signs during this time [16]. In a separate study, systemic lesions associated with viral infection were observed in some pythons without clinical or histologic evidence of respiratory disease [34]. The findings in chameleons could represent similar phenomena as some animals shown to be infected did not develop disease, even over the three months prior to culling for herd health, and exhibited minimal to no histologic lesions (VC7, 8, 9). Other chameleons that were asymptomatic or presented with non-specific clinical signs exhibited inflammatory lesions in the alimentary tract and liver (VC2, 4, 6), which could be an indication of non-respiratory serpentovirus-associated lesions. Future studies are necessary to confirm causation between serpentovirus infection and disease (e.g., in situ diagnostic techniques and experimental infections). Furthermore, infection status over longer periods of time and microscopic or molecular detection of virus in non-respiratory lesions would be necessary to support these claims. 

Bearded dragons, leopard geckos, and ocelot geckos from the same collection were all negative by PCR for VCSV infection. This suggests that either the opportunity for transmission of VCSVs did not occur, transient infection occurred, or these lizards have a reduced susceptibility or resistance to infection with VCSVs. The primers used in this study were designed to specifically target the identified VCSVs, therefore, the results do not rule out the possibility of other serpentovirus infection in these other lizard species. As observed in snakes, divergent serpentoviruses may be present that are not detectable by current targeted diagnostics [16]. Additional testing (e.g., serial PCR or serological evaluation) or metagenomic sequencing from these animals could provide additional insight into the presence or absence of viral infection. 

Coinfection with different virus genotypes appears to be a natural phenomenon for at least some serpentoviruses found in snakes [16]. A recent study in bearded dragons described proliferative lung lesions observed in bearded dragons associated with co-infection by circovirus and parvovirus [3]. In this study, one definitive case of coinfection was identified by metagenomic sequencing and two additional cases of suspected coinfection were identified based on double peaks in Sanger sequencing chromatograms. The chameleon with confirmed coinfection (VC3) died while exhibiting signs of respiratory disease but was not assessed histologically, therefore, the relationship of coinfection and pathologic progression could not be inferred. 

In addition to coinfection with multiple serpentoviruses, a novel orthoreovirus related to known reptilian orthoreoviruses was also detected in some chameleons. Orthoreoviruses are non-enveloped, segmented, double-stranded RNA viruses that have been associated with disease in mammals, birds, and non-avian reptiles [2,35]. In reptiles, including lizards, orthoreoviruses have been found in healthy animals as well as being associated with several disease processes [2,36,37,38,39,40]. However, the link between orthoreovirus infection and disease in lizards, including chameleons, remains circumstantial [41,42]. In contrast to lizards, orthoreoviruses have experimentally been confirmed to cause respiratory disease in some snakes and cause non-pathogenic infection in others [2,43,44]. 

Although we detected coinfection by an orthoreovirus, the available evidence supports the conclusion that serpentoviruses were the primary respiratory pathogens in these chameleons. First, histologic lesions did not support orthoreovirus-induced pathology, which includes syncytial cell formation in areas of proliferative interstitial pneumonia and tracheitis [43]. In this study, syncytial cells were not identified in respiratory lesions or any other tissues of infected chameleons. Second, potential indicators of viral load did not support a direct relationship between orthoreovirus infection and disease. Nearly 3000 orthoreovirus read pairs were detected in VC2 by metagenomic sequencing, suggestive of a higher viral load compared to VC1 and VC3 (less than 40 reads each). However, VC2 was clinically normal, did not have histologic evidence of respiratory disease, had the fewest number of sequencing reads aligning to VCSV (0.1–1% of the number of reads detected in VC1 and VC3), and had a lower Ct by PCR, suggesting a positive correlation between serpentovirus loads and disease and an inverse relationship between orthoreovirus load and disease. Given that the sequencing method used to detect the reovirus sequences was designed to enrich for dsRNA and that this virus was undetectable by qRT-PCR it is difficult to fully grasp the viral load in these chameleons. This does, however highlight the utility of dsRNA-seq for the identification of viruses with dsRNA genomes [20]. The standard shotgun sequencing method we initially used that did not specifically enrich for dsRNA failed to detect the reovirus infection (Table 1). Overall, the lack of unique pathologic features of orthoreovirus infection coupled with the negative correlation of VCOrRV load to disease manifestation suggests that VCOrRV is less likely to be the primary cause of respiratory disease. Reoviruses have occurred as coinfection in other reptiles and have been suggested to be secondary in nature [2,32]. Furthermore, based on serological evidence, orthoreovirus infection in lizards may be commonplace, supporting its role as a potential incidental infection [36,39]. 

Multiple chameleon introductions occurred within the examined collection between 2017 and 2018, including three different “generations” being housed in the same facility and each of these generations comprising several chameleon groups purchased from multiple locations. Between introductions, proper disinfection was not performed and surviving chameleons that had been exposed to diseased animals were subsequently placed with newly arriving stock. Furthermore, newly purchased animals may have experienced transport-associated stress and dehydration, possibly increasing the risk of disease development due to a compromised immune system; this could be especially true in younger animals that were unlikely to be fully immunocompetent. This report emphasizes the importance of adequate disinfection in facilities and quarantine practices, especially those following a disease outbreak of unknown etiology. Infected animals free of clinical signs also represent a potential threat. Clinical and diagnostic assessment of both established and newly arriving animals prior to entry into the collection, as well as implementation of quarantine practices is critical for preventing the persistence and spread of infectious diseases. 

In conclusion, this study describes serpentovirus infection in a collection of veiled chameleons with respiratory disease. Based on the similarities between the disease described in this study and the previous reports of serpentoviruses in lizards and snakes, a causal relationship is considered likely but unproven. The detection of viral RNA by indirect methods (metagenomic sequencing and PCR) without the concomitant use of in situ assays for more direct association of the virus within lesions or the use of experimental infections is insufficient to conclude a definitive association or causation. However, these findings provide a framework for future studies of this virus in chameleons. Furthermore, the number of species affected by serpentoviruses continues to expand, highlighting this class of viruses as an important infectious agent of a diverse and growing set of reptiles. The authors suggest that similar clinical respiratory signs may be caused by serpentoviruses in different squamates and that serpentoviruses should be considered as possible differentials in various species. The significance of coinfection with a novel orthoreovirus is currently unknown, but its potential as a contributing factor in outbreaks of respiratory disease should remain an important consideration for future investigations. 

## Figures and Tables

**Figure 1 viruses-12-01329-f001:**
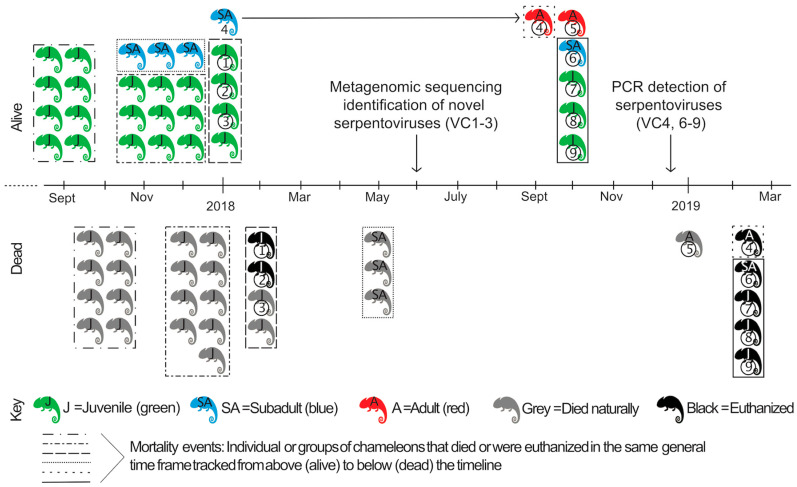
Timeline of case history. A schematic depicting the introduction and death of chameleons into the study collection. Purchased chameleons entering the collection are represented above the timeline (alive). Those same chameleons are represented below the timeline (dead) following euthanasia or natural death; all deaths were associated with clinical signs of respiratory disease except possibly VC5. Individual or mass mortality events are indicated by dashed and dotted lines encircling groups of chameleons. The same border pattern can be found around chameleons above and below the timeline correlating with the groups entry into the collection and then similarly timed deaths. VC4 survived to the age of an adult; progression from subadult to adult is indicated by the horizontal arrow linking the two VC4 symbols. Chameleons that were tested for viral infection (VC1–9) are indicated by the circled identification number directly below the chameleon icon.

**Figure 2 viruses-12-01329-f002:**
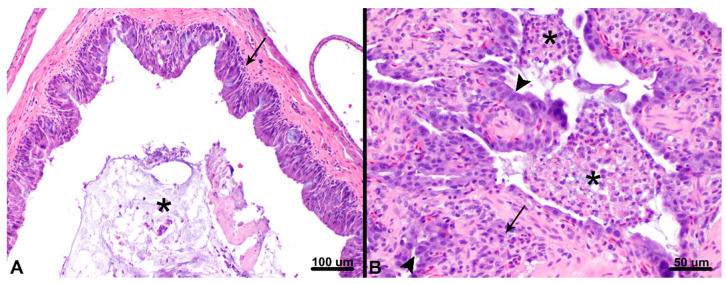
Bronchointerstitial pneumonia and rhinitis with proliferative and catarrhal change in a veiled chameleon (VC1). (**A**) Histopathology of the nasal cavity revealed abundant mucus admixed with low numbers of sloughed epithelial cells, and heterophils (asterisk) with the nasal lumen. Proliferation and lymphocytic infiltration of the nasal mucosa (arrow) was present throughout. Hematoxylin and eosin (HE), 100× magnification, 100 µm scale bar. (**B**) Histopathology of the lungs revealed faveolar lumena (asterisks) and bronchiolar air spaces containing numerous, large accumulations of heterophils admixed with sloughed epithelial cells, mucus and cellular debris. Multifocally, there was marked hypertrophy and hyperplasia of type II pneumocytes (arrowheads) and expansion of interstitial spaces by mononuclear and heterophilic inflammation (arrow). The trachea (not shown) was similarly affected. HE, 200× magnification, 50 µm scale bar.

**Figure 3 viruses-12-01329-f003:**
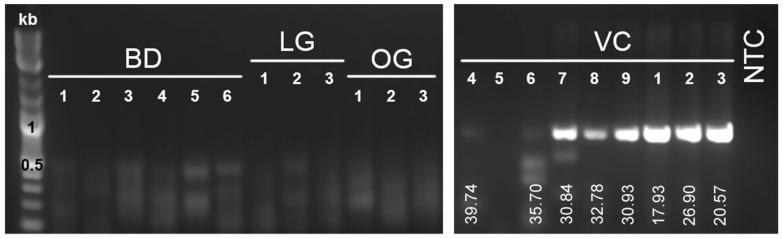
Gel electrophoresis of serpentovirus qPCR products. RNA extracted from tissue homogenates (VC1–3) or oral swabs (VC4–9, BD1–6, LG1–3, and OG1–3) was used in the PCR assay. Expected amplicon size of 752 bp was detected in VC 1–4 and 6–9. Ct values are indicated below each positive band. VC, veiled chameleons. BD, bearded dragons. LG, leopard geckos. OG, ocelot geckos. NTC, no template control (negative).

**Figure 4 viruses-12-01329-f004:**
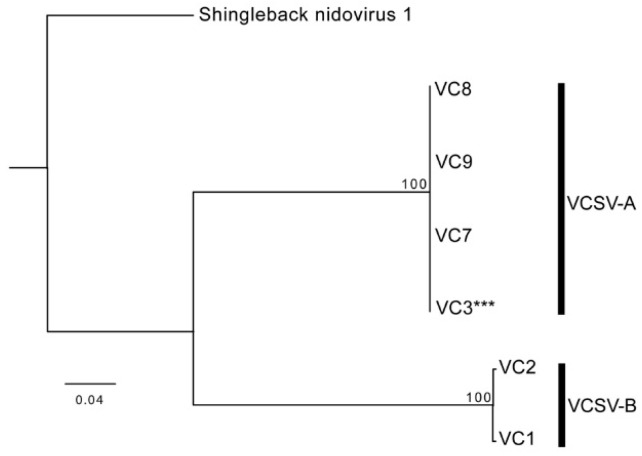
Phylogenetic tree of veiled chameleon serpentoviruses (VCSV). Sequences of PCR amplicons (708 bp) within the ORF1b were aligned and a phylogenetic tree was constructed. Sequences separated into two genotypes: VCSV-A (VC7–9) and VCSV-B (VC1–2). A Sanger sequence of high enough quality could not be obtained for VC3, therefore, the corresponding region derived by metagenomic sequencing (***) of the VCSV-B isolate was utilized in this tree. Maximum likelihood tree constructed using PhyML, LG substitution model, and 1000 bootstrap replicates; bootstrap values (percent) represented at each node. Shingleback nidovirus 1 (AOZ57153.1) was used as the outgroup.

**Figure 5 viruses-12-01329-f005:**
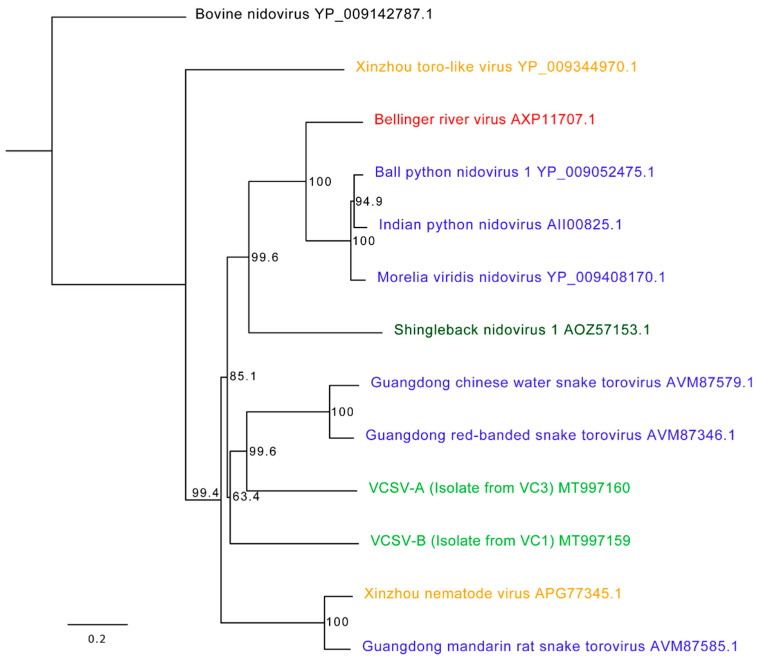
Phylogenetic tree of veiled chameleon serpentoviruses and other reptilian serpentoviruses. The entirety of ORF1b amino acid sequences were aligned and a phylogenetic tree constructed. Veiled chameleon serpentoviruses (VCSV-A and VCSV-B; light green) were compared to known serpentoviruses found in snakes (ball python, Indian python, *Morelia viridis*, Guangdong snake nidoviruses/toroviruses; blue), lizards (shingleback nidovirus; dark green), turtles (Bellinger river virus; red), and snake-associated nematodes (Xinzhou viruses; orange) as well as a remotovirus (bovine nidovirus; black) of the same *Tobaniviridae* family. GenBank accession numbers are included to the right of the virus name. Maximum likelihood tree constructed using PhyML, LG substitution model, and 1000 bootstrap replicates; bootstrap values (percent) represented at each node. Bovine nidovirus outgroup.

**Figure 6 viruses-12-01329-f006:**
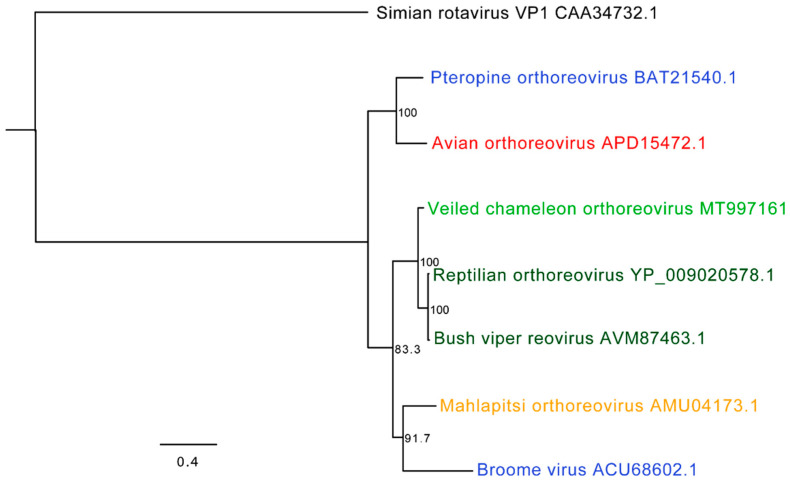
Phylogenetic tree of veiled chameleon orthoreovirus and orthoreoviruses of reptiles, mammals, and birds. The amino acid sequences of L3 segments (which encode the RNA-dependent RNA polymerase) were aligned and a phylogenetic tree constructed. Veiled chameleon orthoreovirus (light green) was compared to known reoviruses found in snakes (reptilian orthoreovirus and bush viper reovirus; dark green), bats (Broome virus and pteropine orthoreovirus; blue), bat flies (Mahlapitsi orthoreovirus; orange), and birds (avian orthoreovirus; red). The VP1 protein (RNA segment 1 encoding the RNA-dependent RNA polymerase) of the Simian rotavirus strain SA11 (black) was used as the outgroup. GenBank accession numbers are included to the right of the virus name. Maximum likelihood tree constructed using PhyML, LG substitution model, and 1000 bootstrap replicates; bootstrap values (percent) represented at each node.

**Table 1 viruses-12-01329-t001:** Summary of sequencing depth and aligning reads to veiled chameleon serpentovirus (VCSV) and veiled chameleon orthoreovirus (VCOrV). The total number of sequences generated per sample. (A) Number of initial reads (single, shotgun RNA; paired, dsRNA). (B) Number of reads remaining after removing low-quality sequences. (C) Number of reads remaining after collapsing non-unique sequences into a single read. (D) Number of reads remaining after removing host-derived sequences. The total number of reads aligning to VCSVs and VCOrV are included.

Sample and Seq Run		Total Reads (A)	Remove Low Quality Reads (B)	Collapse to Unique Reads (C)	Host Filter (D)	VCSV Reads	VCOrV Reads
Shotgun RNA Sequencing	VC1	6.4 × 10^6^	5.7 × 10^6^ (89%)	1 × 10^6^ (15%)	0.5 × 10^6^ (8%)	914	0
	VC2	1.5 × 10^6^	1.4 × 10^6^ (91%)	0.3 × 10^6^ (19%)	0.1 × 10^6^ (9%)	0	0
	VC3	1.1 × 10^6^	1 × 10^6^ (92%)	0.2 × 10^6^ (20%)	0.1 × 10^6^ (10%)	212	0
dsRNA Sequencing	VC1	16.9 × 10^6^	15.7 × 10^6^ (93%)	2.7 × 10^6^ (16%)	2.6 × 10^6^ (15%)	901	39
	VC2	42.7 × 10^6^	38.5 × 10^6^ (90%)	6.6 × 10^6^ (16%)	6 × 10^6^ (14%)	11	2921
	VC3	13.1 × 10^6^	11.4 × 10^6^ (87%)	3.1 × 10^6^ (24%)	3 × 10^6^ (23%)	11,881	11

**Table 2 viruses-12-01329-t002:** Animals sampled for serpentovirus and orthoreovirus testing from a single collection. VCSV, veiled chameleon serpentovirus. VCOrV, veiled chameleon orthoreovirus. M, male. F, female. U, unknown. PM, postmortem. AM, antemortem. LLK, lung-liver-kidney pool. LTOM, lung-trachea-oral mucosa pool. LT, lung-trachea pool. OS, oral swab. MGS, metagenomic sequencing. PCR, polymerase chain reaction. NT, not tested. * Undetected by the methods described but cannot confirm negative infection status.

Species	ID	Sex	Age	Sample Type	Sample Tissue	Analysis	VCSV	VCOrV
Veiled chameleon (*Chameleo calyptratus*)	VC1	M	Juvenile	PM	LLK	MGS, PCR	+	+
	VC2	F	Juvenile	PM	LLK	MGS, PCR	+	+
	VC3	M	Juvenile	PM	LTOM	MGS, PCR	+	+
	VC4	M	Adult	AM, PM	OS, LT	PCR	+	- *
	VC5	F	Adult	AM	OS	PCR	-	- *
	VC6	F	Subadult	AM, PM	OS, LT	PCR	+	- *
	VC7	F	Juvenile	AM, PM	OS, LT	PCR	+	- *
	VC8	F	Juvenile	AM, PM	OS, LT	PCR	+	- *
	VC9	F	Juvenile	AM, PM	OS, LT	PCR	+	- *
Central bearded dragon (*Pogona vitticeps*)	BD1	F	Adult	AM	OS	PCR	-	NT
	BD2	F	Adult	AM	OS	PCR	-	NT
	BD3	M	Adult	AM	OS	PCR	-	NT
	BD4	F	Adult	AM	OS	PCR	-	NT
	BD5	M	Adult	AM	OS	PCR	-	NT
	BD6	F	Adult	AM	OS	PCR	-	NT
Common leopard gecko (*Eublepharis macularius*)	LG1	U	U	AM	OS	PCR	-	NT
	LG2	U	U	AM	OS	PCR	-	NT
	LG3	U	U	AM	OS	PCR	-	NT
Ocelot gecko (*Paroedura pictus*)	OG1	U	U	AM	OS	PCR	-	NT
	OG2	U	U	AM	OS	PCR	-	NT
	OG3	U	U	AM	OS	PCR	-	NT

**Table 3 viruses-12-01329-t003:** General features of the veiled chameleon orthoreovirus genome. The long (L), medium (M), and short (S) encoded genes are indicated. The nucleotide length of each segment is indicated, as is the amino acid length of the encoded protein.

Segment	Gene	Size (bp)	Protein Size (aa)	GenBank Accession
L1	Lambda (λ) A	4000	1131	MT997161
L2	Lambda (λ) C	3685	872	MT997162
L3	Lambda (λ) B (RdRp)	3795	1165	MT997163
M1	Mu (μ) NS	2467	793	MT997164
M2	Mu (μ) A	2289	754	MT997165
M3	Mu (μ) B	2091	613	MT997166
S1	P14 (fusion), Sigma (σ) C	1463	126,350	MT997167
S2	Sigma (σ) A	1260	390	MT997168
S3	Sigma (σ) B	1252	388	MT997169
S4	Sigma (σ) NS	1129	282	MT997170

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
