# Peer review of "Serpentovirus (Nidovirus) and Orthoreovirus Coinfection in Captive Veiled Chameleons (Chamaeleo calyptratus) with Respiratory Disease"

_viruses, 2020, doi:10.3390/v12111329_

Round 1

Reviewer 1 Report

Thank you for the opportunity to review the manuscript entitled "Serpentovirus (nidovirus) and orthoreovirus coinfection associated with respiratory disease in captive veiled chameleons (Chamaeleo calyptratus)"   I found the manuscript to be interesting, well organised and over-all clearly communicated.

Hoon Hanks et al. describe the presentation and epidemiology of multiple mortality events in veiled chameleons where respiratory disease was observed and two novel nidoviruses and a novel orthoreovirus were identified.  Genome sequencing and molecular characterization was performed for each virus. The metatrasncriptomic and genomic processes are thorough and well described. The quality of the manuscript is high and the reported discoveries are a significant contribution to the filed. 

The bulk of my remarks are minor with the following exceptions:

  • figure 4 seems to be missing. 
    The figure legend describes the phylogeny of viral isolates, yet viral culture results were negative.
  • figure 1 is confusing in isolation of the manuscript text.   The figure could be enhanced by including a key to the lines around groups of animals, and by including a description of animal 4 and its associated arrow.  Elevating the sentence "Mass mortality events can be followed by groups of chameleons outlined with similar border patterns" within the figure legend would also assist the reader.
  • figure 2 legend title describes "proliferative and catarrhal interstitial pneumonia".  While the histological description in the remaining figure legend is very good, the title mixes pathological descriptors in a confusing manner.
    The histological description of proliferative and catarrhal change refer to the respiratory epithelium.  As read the title indicates that these changes are happening in the pulmonary interstitium. Proliferative interstitial pneumonia infers an increase in fibroblasts, myofibroblasts or microvasculature, rather than the inflammatory cell  infiltration described here.
    Based on the histological description the reptiles seems to have had broncho-interstitial pneumonia with catarrhal and proliferative change.   Although another pathologist may have other means to accurately pinpoint the location and types of changes occurring in the chameleon lungs.
    Similar clarification is required in the abstract where "proliferative interstitial pneumonia with abundant mucous" is inaccurate as written. 
  • figure 3 is not organised chronologically with the text.  It would benefit the reader to move the chameleon gel electrophoresis image to the left.  The substrate tested (oral swabs, etc) should be included in the figure legend
  • The association between the viruses identified and the pathological changes described is unclear.  The paper describes one animal with concurrent nidovirus infection and histological evidence of significant respiratory lesions.  In situ methods to illustrate the viral agents within lesions, and experimental infection trials have not been conducted to establish that association.  As such the title should be modified to "Serpentovirus (nidovirus) and orthoreovirus coinfection in captive veiled chameleons with respiratory disease" or similar.  The abstract sentence "A causal relationship between serpentovirus infection and respiratory disease in chameleons is considered probable" should also be toned down.   Similarly lines 540-542 should be re-considered in this context.

    The lead author is more cautious in her previous publications, particularly when describing experimental nidovirus infection in ball pythons where previous associations between nidovirus and respiratory disease in that species were charactersed as circumstantial and indirect.  That type of terminology, and a description about the limitations of metatranscriptomics in ascribing agent causality of observed disease would be a meaningful addition to the discussion.

Minor comments regarding the remainder of the text include:

  • line 50 - unusual space and character in discovered
  • lines 64 and 126-127  how was the case history suggestive of infectious etiology rather than intoxication, husbandry-associated illness, or other disease processes?
  • line 132 - : and feces were submitted
  • line 163 - perhaps more precise to say PCR testing rather than virological.  It may be more accurate to remove the word also from this sentence given that these were the only samples collected from the animals.
  • line 177 - according to manufacturers instructions
  • line 206 was added
  • line 262 - as above the term "catarrhal interstitial pneumonia" is confusing.  Perhaps "proliferative and catarrhal broncho-interstitial pneumonia"
  • line 298 - diagnostics - jargon, diagnostic techniques or investigations
  • line 307 (VCSV-A and B)
  • Table 1 and 2 - it might be helpful for VCSV and VCOrRV to be defined in the title or legend, also in figure legends.
  • line 377 - the word thus is out of place based on the context set by the first portion of the sentence.
  • line 466 - as above regarding the use of "association
  • line 454 - indicating a similar pathogenesis seems strong.  Suggesting may be a more approriate term.
  • line 462-464 - although age may affect animal susceptibility to disease, other potential factors contributing to disease prevalence and expression should be included in the discussion (eg infectious dose, mode of transmission, concurrent infection, nutritional and immunological status)
  • line 464 and 473 - "and/or" is vague and takes the reader's time to understand intended meaning
  • lines 478-480 - experimental infection studies and in-situ diagnostic techniques should be included in the process required to further understand the link between VCSVs and respiratory disease.
  • line 483 - transient infection of co-housed reptiles can not be ruled out based on the data presented.
  • line 487-8 - Metatranscriptomics is a pretty expensive tool to be deployed for surveillance of populations, but it could augment serial PCR testing or serological testing.
  • line 496-7 - also detected in some chameleons. indicating coinfection with multiple viruses. (redundant).
  • line 502 - confirmed to cause of respiratory disease

The  discussion may benefit from the inclusion of a recent description of proliferative lung lesions observed in bearded dragons found to be infected with circovirus and Chaphamaparvovirus, recently published in Viruses.

Thank you again for the opportunity to read this manuscript.

Author Response

Reviewer 1

Comments and Responses:

Major Comments:

1.Figure 4 seems to be missing.The figure legend describes the phylogeny of viral isolates, yet viral culture results were negative.

We inadvertently omitted Figure 4 but it is now included; see Major Comment 4 for placement notes. The authors agree that “isolates” was the wrong terminology to use and it has been removed.

2.Figure 1 is confusing in isolation of the manuscript text.

The figure could be enhanced by including a key to the lines around groups of animals, and by including a description of animal 4 and its associated arrow.

Elevating the sentence "Mass mortality events can be followed by groups of chameleons outlined with similar border patterns" within the figure legend would also assist the reader.

To increase the stand-alone understandability of the figure, we added the following in-figure text associated with dashed line legend: “Mortality events: Individual or groups of chameleons that died or were euthanized in the same general time frame tracked from above (alive) to below (dead) the timeline.”We’ve also edited the figure legend to reflect the changes requested by the reviewer. This now reads: “Figure 1. Timeline of case history. A schematic depicting the introduction and death of of chameleons into the study collection. Purchased chameleons entering the collection are represented above the timeline (alive). Those same chameleons are represented below the timeline (dead) following euthanasia or natural death; all deaths were associated with clinical signs of respiratory disease except possibly VC5. Individual or mass mortality events are indicated by dashed and dotted lines encircling groups of chameleons. The same border pattern can be found around chameleons above and below the timeline correlating with the groups entry into the collection and then similarly timed deaths. VC4 survived to the age of an adult;

progression from subadult to adult is indicated by the horizontal arrow linking the two VC4 symbols. Chameleons that were tested for viral infection (VC1-9) are indicated by the circled identification number directly below the chameleon icon..”

3.Figure 2 legend title describes "proliferative and catarrhal interstitial pneumonia".

While the histological description in the remaining figure legend is very good, the title mixes pathological descriptors in a confusing manner.

The histological description of proliferative and catarrhal change refer to the respiratory epithelium.

As read the title indicates that these changes are happening in the pulmonary interstitium. Proliferative interstitial pneumonia infers an increase in fibroblasts, myofibroblasts or microvasculature, rather than the inflammatory cell infiltration described here.

Based on the histological description the reptiles seems to have had broncho-interstitial pneumonia with catarrhal and proliferative change.

Although another pathologist may have other means to accurately pinpoint the location and types of changes occurring in the chameleon lungs.

Similar clarification is required in the abstract where "proliferative interstitial pneumonia with abundant mucous" is inaccurate as written.

The authors understand the confusion with the term “proliferative and catarrhal interstitial pneumonia.” Proliferative pneumonia or proliferative interstitial pneumonia have been diagnoses commonly used to describe pneumonia in reptiles that include proliferative changes of the epithelium, especially in the context of viral infection.

These diagnoses have been repeatedly used in previous descriptions of serpentovirus infection in reptiles by multiple groups.

However, the authors agree that a more explicit description of the lesions could be used. The Figure 2 title has been changed to read “Bronchointerstitial pneumonia and rhinitis with proliferative and catarrhal change in a veiled chameleon (VC1)”.The abstract has been changed to read “Histopathologic lesions included rhinitis and interstitial pneumonia with epithelial proliferation and abundant mucus”.

4.Figure 3 is not organized chronologically with the text.

It would benefit the reader to move the chameleon gel electrophoresis image to the left.

The substrate tested (oral swabs, etc) should be included in the figure legend.

Figure 3 and Figure 4 are initially referenced in the first paragraph of section 3.3 Targeted Viral RNA Detection. Figures 3 and 4 have been placed directly after this paragraph. These figures are subsequently referenced multiple times in the second paragraph of this section as well. In the Figure 3 legend, the substrate has been added by the inclusion of the following sentence: “RNA extracted from tissue homogenates (VC1-3) or oral swabs (VC4-9, BD1-6, LG1-3, and OG1-3) was used in the PCR assay”.

5.The association between the viruses identified and the pathological changes described is unclear.

The paper describes one animal with concurrent nidovirus infection and histological evidence of significant respiratory lesions.In situ methods to illustrate the viral agents within lesions, and experimental infection trials have not been conducted to establish that association.

As such the title should be modified to "Serpentovirus (nidovirus) and orthoreovirus coinfection in captive veiled chameleons with respiratory disease" or similar.

The abstract sentence "A causal relationship between serpentovirus infection and respiratory disease in chameleons is considered probable" should also be toned down.

Similarly lines 540-542 should be re-considered in this context.

The lead author is more cautious in her previous publications, particularly when describing

experimental nidovirus infection in ball pythons where previous associations between nidovirus and respiratory disease in that species were characterized as circumstantial and indirect.

That type of terminology, and a description about the limitations of metatranscriptomics in ascribing agent causality of observed disease would be a meaningful addition to the discussion.

The authors agree that the language could be toned down. The methods used in this study limit the conclusions that can be drawn regarding causation. The title has been changed to “Serpentovirus (nidovirus) and orthoreovirus coinfection in captive veiled chameleons (Chamaeleo calyptratus) with respiratory disease”.

The relevant wording in the abstract has been changed from “considered probable” to “suspected”. The discussion has been altered and an additional statement has been added to reflect the reviewers comments: “Based on the similarities between the disease described in this study and the previous reports of serpentoviruses in lizards and snakes, a causal relationship is considered likely but unproven. The detection of viral RNA by indirect methods (metagenomic sequencing and PCR) without the concomitant use of in situ assays for more direct association of the virus within lesions or the use of experimental infections is insufficient to conclude a definitive association or causation. However, these findings provide a framework for future studies of this virus in chameleons.”

Minor Comments:

1.Line 50 -unusual space and character in discovered

This has been fixed.

2.Lines 64 and 126-127howwas the case history suggestive of infectious etiology rather than intoxication, husbandry-associated illness, or other disease processes?

Multiple factors suggested an infectious etiology could be contributing to the disease (although it wasn’t the only differential). The fact that juveniles tended to be more severely affected(possible age-associated immunosuppression), that other lizard species housed within the same area were unaffected, and multiple attempts at changing husbandry practices and environmental factors did not have significant effects. Multiple avenues were pursued by the collection owner to try and mitigate morbidity and mortality and an exhaustive list did not seem necessary to include. The introduction sentence has been modified: “Aspects of the case history were suggestive of a possible infectious etiology, but traditional diagnostic methods did not reveal a potential bacterial, fungal, or viral agent”. The case history sentence has been modified: “The case history, clinical findings, and postmortem findings (see below) suggested an infectious agent could be playing a role”.

3.Line 132 -:and feces were submitted

This has been addressed.4.Line 163 -perhaps more precise to say PCR testing rather than virological. It may be more accurate to remove the word also from this sentence given that these were the only samples collected from the animals. The word virological has been removed from the sentence.5.Line 177 -according to manufacturers instructions

This has been fixed.6.Line 206wasaddedThis has been corrected

7.Line 262 -as above the term "catarrhal interstitial pneumonia" is confusing. Perhaps "proliferative and catarrhal broncho-interstitial pneumonia "The sentence has been changed to “chronic-active bronchointerstitial pneumonia and tracheitis with proliferative and catarrhal changes”

8.Line 298 -diagnostics -jargon, diagnostic techniques or investigations

The sentence has been changed to include “diagnostic techniques”

9.Line 307 (VCSV-A and B)The sentence has been changed to “The same serpentoviruses were detected in VC1 (VCSV-B) and VC3 (VCSV-A and B)”.

10.Table 1 and 2 -it might be helpful for VCSV and VCOrRV to be defined in the title or legend, also in figure legends. Table 1 title has been updated: Summary of sequencing depth and aligning reads to veiled chameleon serpentovirus (VCSV) and veiled chameleon orthoreovirus (VCOrV)Table 2 has been updated to include: VCSV, veiled chameleon serpentovirus. VCOrV, veiled chameleon orthoreovirus.

Figure 4 title has been updated: Phylogenetic tree of veiled chameleon serpentoviruses (VCSV).

11.Line 377 -the wordthusis out of place based on the context set by the first portion of the sentence.The word thus has been removed from the sentence.

12.Line 466 -as above regarding the use of "association"The word association was not found in this line or other lines in the area. However, any references in the manuscript to “serpentovirus associated with respiratory disease in chameleons” has been changed to “serpentovirus in chameleons with respiratory disease”(see also Major Comment 5 response).

13.Line 454 -indicating a similar pathogenesis seems strong. Suggesting may be a more appropriate term.

This has been changed to “suggesting”.

14.Line 462-464 -although age may affect animal susceptibility to disease, other potential factors contributing to disease prevalence and expression should be included in the discussion (eg infectious dose, mode of transmission, concurrent infection, nutritional and immunological status)

The authors agree that other factors of susceptibility could be at play, but an extensive discussion would continue to be speculation. A sentence has been added to include other possible factors: “In this study, the opposite trend was observed, suggesting that physiologic differences, such as incomplete immune development, could contribute to decreasing susceptibility with increasing age, although this hypothesis will require additional investigation. Other possible factors that could contribute to variation in susceptibility include infectious dose, mode of transmission, concurrent infection or comorbidities, nutritional status, stress response, or other immunologic factors.”

15.Line 464 and 473 -"and/or" is vague and takes the reader's time to understand intended meaning

This sentence was modified: “Serpentoviruseswere detected in chameleons exhibiting clinical signs of respiratory disease with or without histologic correlates, as well as clinically healthy animals lacking microscopic disease within the lower respiratory tract.”

16.Lines 478-480 -experimental infection studies and in-situ diagnostic techniques should be included in the process required to further understand the link between VCSVs and respiratory disease.

The section has been modified to reflect the additional diagnostics needed: “Future studies are necessary to confirm causation between serpentovirus infection and disease (e.g. in situ diagnostic techniques and experimental infections). Furthermore, infection status over longer periods of time and microscopic or molecular detection of virus in non-respiratory lesions would be necessary to support these claims.”

17.Line 483 -transient infection of co-housed reptiles cannot be ruled out based on the data presented.

The sentence has been modified: “This suggests that either the opportunity for transmission of VCSVs did not occur, transient infection occurred, or these lizards have a reduced susceptibility or resistance to infection with VCSVs.”

18.Line 487-8 -Metatranscriptomics is a pretty expensive tool to be deployed for surveillance of populations, but it could augment serial PCR testing or serological testing.

This sentence has been updated: “Additional testing (e.g. serial PCR or serological evaluation) ormetagenomic sequencing from these animals could provide additional insight into the presence or absence of viral infection.”

19.Line 496-7 -also detected in some chameleons indicating coinfection with multiple viruses. (redundant).

This has been removed

20.Line 502 -confirmed to cause of respiratory disease

This has been addressed

21.Thediscussionmay benefit from the inclusion of a recent description of proliferative lung lesions observed in bearded dragons found to be infected with circovirus and Chaphamaparvovirus, recently published in Viruses.

We cited the recent publication titled “Meta-transcriptomic discovery of a divergent circovirus and a chaphamaparvovirus in captive reptiles with proliferative respiratory syndrome”as an excellent example of the use of metatranscriptomic/metagenomic sequencing technology in the discovery of novel pathogens in the first sentence of the introduction and as an example of coinfection associated with lung disease in the discussion paragraph that addresses co-infection: “A recent study in bearded dragons described proliferative lung lesions observed in bearded dragons associated with co-infection by circovirus and parvovirus.”

Reviewer 2 Report

Author 8 (Hailey Sten) is cited as author 9

Author Response

Editor Comments and Responses:

1.When we processed your manuscript we were not able to find the number of theapproval from the Ethic Committee. Please provide it and insert it in yourmanuscript when revising your paper.

An ethics statement has been added to the Materials and Methods (Section 2.1; Lines 79-88): “The collection of lizards described below was enrolled in a separate biological research project for which IACUC review was independently completed (Southeastern Louisiana University IACUC Protocol #0046). For the study described in this report, all antemortem samples were considered non-invasive clinical samplescollected by veterinarians for diagnostic purposes. Postmortem samples were also collected for diagnostic purposes. Euthanasia of animals was performed by a veterinarian for the purposes of herd health; use of diagnostic samples for this study was considered secondary and did not influence clinical decisions. For these reasons an ethical review for this study was not performed.”

Additionally, following further discussion amongst the authors, an individual referenced in the acknowledgements section has been changed to an author (Raúl E. Díaz). His contact information has been added.

Reviewer 2

Comments and Responses:

1.Author 8 (Hailey Sten) is cited as author 9

This has been addressed and all author numbers are now correctly cited
